# Altered Mitochondrial Morphology and Bioenergetics in a New Yeast Model Expressing Aβ42

**DOI:** 10.3390/ijms24020900

**Published:** 2023-01-04

**Authors:** Khoren K. Epremyan, Anton G. Rogov, Tatyana N. Goleva, Svetlana V. Lavrushkina, Roman A. Zinovkin, Renata A. Zvyagilskaya

**Affiliations:** 1A.N. Bach Institute of Biochemistry, Research Center of Biotechnology of the Russian Academy of Sciences, Leninsky Ave. 33/2, 119071 Moscow, Russia; 2National Research Center “Kurchatov Institute”, Akademika Kurchatova pl. 1, 123182 Moscow, Russia; 3Belozersky Institute of Physico-Chemical Biology, Lomonosov Moscow State University, Leninskye Gory 1/40, 119992 Moscow, Russia; 4Faculty of Bioengineering and Bioinformatics, Lomonosov Moscow State University, Leninskye Gory 1/73, 119234 Moscow, Russia; 5The “Russian Clinical and Research Center of Gerontology” of the Ministry of Healthcare of the Russian Federation, Pirogov Russian National Research Medical University, Ostrovityanova st. 1, 117997 Moscow, Russia

**Keywords:** Alzheimer’s disease, Aβ42, heterologous expression, yeast, *Yarrowia lipolytica*, cell death, oxidative stress, mitochondrial dysfunction, mitochondrial fragmentation, SkQThy

## Abstract

Alzheimer’s disease (AD) is an incurable, age-related neurological disorder, the most common form of dementia. Considering that AD is a multifactorial complex disease, simplified experimental models are required for its analysis. For this purpose, genetically modified *Yarrowia lipolytica* yeast strains expressing Aβ42 (the main biomarker of AD), eGFP-Aβ42, Aβ40, and eGFP-Aβ40 were constructed and examined. In contrast to the cells expressing eGFP and eGFP-Aβ40, retaining “normal” mitochondrial reticulum, eGFP-Aβ42 cells possessed a disturbed mitochondrial reticulum with fragmented mitochondria; this was partially restored by preincubation with a mitochondria-targeted antioxidant SkQThy. Aβ42 expression also elevated ROS production and cell death; low concentrations of SkQThy mitigated these effects. Aβ42 expression caused mitochondrial dysfunction as inferred from a loose coupling of respiration and phosphorylation, the decreased level of ATP production, and the enhanced rate of hydrogen peroxide formation. Therefore, we have obtained the same results described for other AD models. Based on an analysis of these and earlier data, we suggest that the mitochondrial fragmentation might be a biomarker of the earliest preclinical stage of AD with an effective therapy based on mitochondria- targeted antioxidants. The simple yeast model constructed can be a useful platform for the rapid screening of such compounds.

## 1. Introduction

Two forms of Alzheimer’s disease (AD) [1] are known: the early onset familial (accounting 1–2% of all AD cases) and the most common late-onset sporadic form. The sporadic form is a progressive, complex, incurable age-dependent neurodegenerative disorder, causing the predominant form of dementia marked by the progressive loss of neuronal structure, multiple cognitive impairments, and changes in behavior and personality [2,3,4,5].

AD is currently ruining the lives of over 50 million people [2,6] and this number is expected to rise significantly as a result of increased life expectancy [2,7,8,9]. With no current treatments to prevent or reverse AD, the cumulative cost of care for AD patients will place a heavy burden on health care systems worldwide [8]. 

Some risk factors can cause AD, with aging and immunosenescence being the leading ones [10]. Individuals with the ε4 allele of apolipoprotein E (APOE4) and mutations in other cholesterol transporters in the brain have a higher chance of the disease [11,12,13]. Modifiable risk factors of AD and dementia include alterations in the blood–brain barrier [14,15,16], vascular disorders (increased blood pressure) [11,12,17], metabolic factors (increased insulin resistance in type two diabetes mellitus, obesity, increased cholesterol content, impaired glucose metabolism) [12,17,18], infections, inflammation [15,19], gender (female) predominance [20], lifestyle factors (unhealthy diet, alcohol misuse, smoking, physical inactivity, low educational attainment in early life, dysbiosis) [11,12,21]. Other causes include family history [22], stress, depression, inadequate sleep, and infrequent social contacts [17].

It is generally accepted that the deposition of extracellular senile plaques of insoluble β-amyloid peptide (Aβ42) and intraneuronal inclusions (neurofibrillary tangles, NFT) composed of truncated and hyperphosphorylated forms of the microtubule-stabilizing protein tau (P-tau), loss of synapses and neurons, and changes in the morphology and function of microglia and astrocytes are the most pathologically important phenotypic hallmarks of AD [8,15,23,24].

The amyloid cascade hypothesis, which dominated for a long time [25,26], postulated that Aβ aggregates initiate a cascade of cellular changes including tau protein hyperphosphorylation, inflammation, oxidative mitochondrial damage, mitochondrial deficiency, release of pro-apoptotic factors, neuronal apoptosis causing synaptic failure and neuronal loss, neurotoxicity, and dementia [27,28]. However, the validity of the hypothesis was increasingly questioned [29,30,31]. Moreover, growing evidence has shown that the main pathological processes had already taken place decades before the first symptoms were clinically diagnosed [32]. Novel mechanisms and new hypotheses have been proposed to explain the development and progression of this disease including oxidative stress hypothesis, cell cycle hypothesis, vascular and cholesterol hypothesis, the hypothesis of impaired insulin signaling, and others [33]. Recently, mitochondrial dysfunction has been considered as one of the earliest intracellular processes involved in many neurodegenerative diseases including AD [5,34,35].

With that in mind, the NIA-AA (National Institute on Aging and Alzheimer’s Association) has recommended considering AD as a disease marked by a gradual progression with dementia at its final stage and to diagnose AD by monitoring biomarkers, objectively measurable parameters underlying the pathology in patients during life, using clinical symptoms only to stage of the disease [27,36,37]. Timely detection of the early stage of AD is especially crucial, as it could facilitate the implementation of early and hence efficient treatments of this disorder. An accurate early diagnosis of AD by using sensitive, specific, reliable, reproducible, noninvasive, and simple to perform biomarkers to detect specific AD pathology and to select optimal patient care is central in AD research [38,39]. 

Today, biomarkers for the detection of preclinical AD progression include: ratio Aβ42:Aβ40, total tau (T-tau), and phosphorylated tau (P-tau) in cerebrospinal fluid (CSF) [40,41], measured using positron emission tomography (PET) with tracers specific for Aβ, tau, and synaptic impairment [42]. However, the global use of use of CSF and PET is not widespread due to high costs and invasiveness [43]. The newer models for the early diagnosis of AD suggest using saliva [44], the eye [45], and blood [37,46].

The quest for a better understanding of the molecular mechanisms underlying AD, with its variety of symptoms and complicated cross-talk of cofactors, has led to the development of more simple eukaryotic models including animal models (reviewed in [47,48,49,50,51]), *C. elegans* (reviewed in [52,53,54]), and *Drosophila* (reviewed in [53,55,56]).

The field of AD research has greatly benefited from the use of these models, especially in understanding the mechanisms of Aβ and P-tau aggregation and the loss of their functions. Despite these scientific advances, however, these models remained too complex and did not fully reproduce the human AD pathology.

The search for a simple model made researchers pay attention to well-defined unicellular yeasts, the simplest eukaryotic organisms. The high degree of preservation of various fundamental biological processes including cell growth and division, organelle function, energy metabolism, proteostasis, signal transduction, stress responses, vesicle trafficking, cell cycle progression, endocytosis, aging, and cell death [57] makes yeast a promising model for AD research [58]. Moreover, 31% of yeast genes have human orthologues [59], lending additional support to possible functional discovery investigations using this model. 

Yeast offers greater advantages over other disease models, allowing for a rapid and relatively easy method of establishing gene–protein–function associations, mainly due to the simplicity, short life cycle, low-cost cultivation techniques giving large number of genetically homogeneous cells, susceptibility to simple genetic and environmental manipulations, and the accessibility of powerful genomic and proteomic tools and high-throughput screening techniques, thus strengthening the rationale for using yeast as a valuable AD model [57,60].

However, despite being a useful and powerful model system, yeasts have their natural limitations. Cell–cell interactions, synaptic transmissions, axonal transport, glial–neuronal interactions, immune and inflammatory responses, neuronal specializations play an important role in neurodegeneration, and the cognitive aspects of AD cannot be reproduced in yeast. Nevertheless, despite these limitations, yeast models play an increasingly important role in revealing the main fundamental processes in AD and the screening of AD-related compounds [61].

Although *S. cerevisiae* is the most common model, this facultative anaerobe with less abundant, small-sized, poorly structured mitochondria is not a bioenergy equivalent of high-energy demanding neurons, relying almost exclusively on mitochondrial oxidative phosphorylation. In this respect, *Yarrowia lipolytica*, a non-toxic aerobic “multitalented” yeast species with a well-characterized genome, having GRAS (generally regarded as safe) status, metabolic diversity and flexibility, rapid utilization rates, unique biosynthetic and secreting capacities, energy metabolism largely reminiscent of that in mammals and susceptible to molecular genetic engineering tools (reviewed in [62]), may be a highly promising alternative model for decoding mitochondria-related AD pathogenesis.

Moreover, the Po1f strain of *Y. lipolytica* used in the work is auxotrophic for uracil and leucine and has a deletion of the Xpr2 gene encoding an extracellular protease, thus allowing cells to grow on selective media.

Thus, the main goal of the work was to create, to our knowledge, for the first time, an improved *Y. lipolytica*-based yeast model to detect the direct effect of Aβ42 amyloid expression on the mitochondrial structure and dynamics, the redox status, and viability of cells as well as on the bioenergetics at the mitochondrial level.

## 2. Results

### 2.1. Development and Primary Characterization of Y. lipolytica Cells Expressing Aβ42

*Y. lipolytica* cells expressing Aβ42 and Aβ40 peptides containing 42 and 40 C-terminal amino acid residues of beta-amyloid, respectively, as well as their fusion-constructs eGFP-Aβ42 and eGFP-Aβ40 proteins linked with GFP at the N-terminus were developed. The control *Y. lipolytica* Po1f pZ-0 strain, not having the target proteins but carrying the pZexpress++ integrative plasmid containing the URA3 gene as a prototrophic factor as well as the Po1f pZ-eGFP strain were also constructed. Cultivation of mutants demonstrated their viability. Activity of the hp4d promoter was optimized; the high expression level was determined using the Po1f pZ-eGFP strain by measuring the intensity of green fluorescence by flow cytometry.

### 2.2. Morphology of Mitochondria in Yeast Cells

Stained *Y. lipolytica* cells were visualized using structured illumination microscopy (SIM), providing super-high resolution images of the objects under study. The control (pZ-0) strain contained the branched mitochondrial reticulum (Figure 1, top left panel).

In contrast, the mitochondrial reticulum was disturbed in the pZ-Aβ42 and pZ-eGFP-Aβ42 cells and the mitochondria were fragmented (Figure 1, bottom left panel and right panel). These cells were also marked by the presence of one or more large aggregates and several small aggregates as well as fragmented mitochondria diffusely located in the cell. Similar results were obtained upon the treatment of the pZ-Aβ42 cells with antibodies to the Aβ42 protein (Figure 1, bottom left panel), collectively indicating the strong negative impact of Aβ42 on the mitochondrial structure.

For comparative analysis of the mitochondrial morphology in the Po1f pZ-eGFP-Aβ42 and Po1f pZ-eGFP-Aβ40 mutants, we used another super-resolution method—widefield fluorescence microscopy (Figure 2).

Po1f pZ-eGFP cells retained the mitochondrial reticulum and eGFP was diffusely distributed in the cytosol (Figure 2, left panel). In full accordance with the data presented in Figure 1, the Po1f pZ-eGFP-Aβ42 mutant was characterized by fragmented mitochondria and Aβ42 aggregation, as inferred from compact pronounced eGFP fluorescence (Figure 2, left panel). The Po1f pZ-eGFP-Aβ40 mutant had a normal branched mitochondrial reticulum and diffuse intracellular eGFP-related fluorescence similar to the Po1f pZ-eGFP control, suggesting that the expression of eGFP-Aβ40 does not have any significant effect on the mitochondrial structure. Since the data in Figure 1 and Figure 2 were fully consistent with those from other models, it can be concluded that we actually accomplished the main task of constructing (developing) a simple yeast model, allowing us to study the direct effects of amyloid proteins on the mitochondrial structure. Moreover, we were able to show that the preincubation of cells with very low (250 nM) concentrations of the effective mitochondria-targeted lipophilic antioxidant SkQThy did not affect the mitochondrial reticulum in the control (Po1f and pZ-eGFP) and pZ-eGFP-Aβ40 expressing cells, while largely restoring the mitochondrial reticulum in cells expressing Aβ42 (Figure 2, right panel).

### 2.3. Assessment of Oxidative Status and Viability of Y. lipolytica Cells

The oxidative status and viability of *Y. lipolytica* yeast cells were evaluated using flow cytometry by double-staining with dihydroethidium (DHE) and Sytox Green. Three cell populations could be revealed according to the level of dye fluorescence: living cells not subjected to oxidative stress were marked by a low level of fluorescence of both dyes; living cells experiencing oxidative stress were characterized by high DHE and low Sytox Green fluorescence and dead cells with a high level of fluorescence for both dyes (Figure 3A).

The Aβ42-expressing cells experienced stronger oxidative stress than the control cells. Preincubation of these strains with 250 nM SkQThy for 1 h noticeably decreased oxidative stress and significantly increased the viability of the cells. Incubation for 2 h of both strains with t-BHP [63] expectedly increased oxidative stress while decreasing the viability. When preincubation with SkQThy preceded the treatment with the prooxidant, the positive effect of SkQThy was noticeable and reproducible.

The most prominent and promising effect of SkQThy was almost total elimination of death in Aβ42-expressing cells, which aligned very well with the substantial recovery of the mitochondrial reticulum in cells expressing Aβ42 (Figure 2, right panel).

### 2.4. Characterization of Mitochondria Isolated from Yeast Cells

For a better understanding of the impact of Aβ42 expression on the bioenergetics of cells, we compared the energy parameters of mitochondrial preparations isolated from the control Po1f strain and pZ-Aβ42 cells (Figure 4A,B). Mitochondria from the control cells were tightly coupled, and did not differ from the mitochondrial preparations usually obtained in previous studies [64]. In contrast, mitochondrial preparations from pZ-Aβ42 cells were loosely coupled (partially uncoupled), as inferred from the lowered respiratory control ratios (Figure 4B).

The classic uncoupler carbonylcyanide *m*-chlorophenylhydrazone (CCCP) enhanced mitochondrial respiration in state 4 (Figure 5), with a maximal effect attained at a concentration of 300 nM for mitochondria from both the control cells and cells expressing Aβ42. However, the efficiency of CCCP uncoupling varied significantly, being much more pronounced for mitochondria from the control cells. Since, under optimal conditions, the uncoupling effect of CCCP is usually greater than the measured respiratory control value (the degree of coupling respiration with phosphorylation), it is generally accepted that CCCP-enhanced respiration may serve as an indicator of the maximal possible respiratory control value. With these premises in mind, these data reinforce the notion that mitochondria from Aβ42 expressing cells were partially uncoupled.

Nevertheless, being partially uncoupled, mitochondria from the Aβ42 expressing cells retained an intact structure of the respiratory chain, as shown from the analysis of respiratory titration curves by specific inhibitors (i.e., rotenone (for Complex I) (Figure 6A) and antimycin A (AA, for Complex III) (Figure 6B)). Thus, Aβ42 expression most likely does not disturb the functioning of complexes I and III of the respiratory chain.

The ATP production by the mitochondria of cells expressing Aβ42 was on average 25% lower than in the control strain (Figure 7). Oligomycin, the specific inhibitor of ATP synthase, was used as a negative control. The differences between the two strains were small, but reliable. In mitochondria from the control strain, oligomycin almost completely inhibited ATP synthesis, while in mitochondria from the Aβ42 expressing cells, the degree of inhibition was reliably lower, again indicating that these mitochondria were partially uncoupled.

Finally, hydrogen peroxide production by mitochondrial preparations isolated from the control strain and the Aβ42 expressing cells was measured fluorometrically by changing the fluorescence level of resorufin. To attain the maximal hydrogen peroxide production, mitochondrial catalase (decomposing hydrogen peroxide) was inhibited by aminotriazole. It was found that in mitochondria from the cells expressing Aβ42, the rate of hydrogen peroxide production was higher than in the control counterparts (Figure 8). The prooxidant t-BHP enhanced the production of hydrogen peroxide by mitochondria from both strains, with mitochondria from the Aβ42-expressing cells being more sensitive to oxidative stress than mitochondria from the control strain.

Thus, the comprehensive comparative study of energy parameters of mitochondria isolated from the control strain and Aβ42-expressing cells showed that Aβ42 expression induced mitochondrial dysfunction, as evident from the partial uncoupling of respiration and phosphorylation, the reduced ATP production rate, and the increased susceptibility to oxidative stress.

## 3. Discussion

AD is now considered as a long-developing process, with the onset of both amyloid plaques and NFT accumulation one to two decades before the first cognitive symptoms emerge. This preclinical phase is the best time for AD prevention. 

Currently, mitochondria-targeted therapies of AD are receiving increasing attention [65]. Mitochondria are critical for cell survival and death. In addition to their well-known energy storage function, they are deeply integrated into the cellular metabolism, required for growth and proliferation; implicated in modulating cell signaling, maintaining redox homeostasis, and regulating cell adaption to external stress factors and apoptosis (reviewed in [63]). To fulfill their energy needs, neurons, post-mitotic and excitable cells, rely almost exclusively on the mitochondrial oxidative phosphorylation system. There is a common view that mitochondria are the initial and the principal organelles responsible for ROS generation in the cell. Moreover, as believed, it is excessive mitochondrial ROS production as a result of mitochondrial dysfunction that leads to the development of many pathologies including AD [66,67]. Analysis of the different thus far described models have shown that AD is characterized by mitochondrial dysfunction, as inferred from the decreased mitochondrial membrane potential, respiration and ATP/ADP ratio, impaired mitochondrial trafficking in neurons [65,68], excessive mitochondrial fragmentation [69,70,71], and enhanced ROS production as a result of an imbalance between ROS formation and clearance [72,73]. 

Importantly, we obtained almost the same results using the newly developed simplest yeast model based on *Y. lipolytica* cells expressing Aβ42. The mitochondrial dysfunction was manifested by the loose coupling of respiration and phosphorylation (Figure 4, Figure 5 and Figure 6), the reduced ATP production rate (Figure 7), increased ROS production (Figure 3A,B and Figure 8), and the reduced resistance to oxidative stress (Figure 8). By using two super-resolution imaging methods—SIM (Figure 1) and widefield (Figure 2) fluorescent microscopy—it was shown that in pZ-eGFP-Aβ42 cells, in contrast to the control cells, the mitochondrial reticulum was disturbed and mitochondria were fragmented. These cells were also marked by the presence of Aβ42 aggregates in the cell. Similar results were obtained upon the treatment of pZ-Aβ42 cells with antibodies to Aβ42 protein (Figure 1), thus collectively indicating the strong negative impact of Aβ42 on the mitochondrial structure. Moreover, in accordance with the theoretical predictions, the yeast cells expressing Aβ40 retained the mitochondrial reticulum and were practically identical in this respect to the control cells expressing only GFP; no mitochondrial fragmentation was evident (Figure 2). Additionally, it was clearly demonstrated that low (submicromolar) concentrations of SkQThy, the most effective mitochondria-directed antioxidant in the SkQ-family [61], almost completely prevented mitochondrial fragmentation (Figure 2) and reduced oxidative stress (Figure 3A,B).

These data highlight the utility and desirability of using yeast models to identify mitochondrial dysfunction underlying AD. Equally important is that SkQThy can be useful for mitigating or even preventing harmful effects in AD, and the developed *Y. lipolytica* model provides a promising platform for the rapid screening of such effective compounds. 

Recently, interest in mitochondrial-directed antioxidants has increased dramatically [66,72,74,75,76]. Lipophilic, mitochondria-targeted cationic drugs (predominantly, if not exclusively, transported to and accumulated into the negatively charged mitochondria), especially linked via saturated hydrocarbon chain to a quinone or plastoquinone moiety possessing a strong antioxidant activity, are the most efficient mitochondria-targeted antioxidants [76,77]. As they are transported into mitochondria in conformity with the membrane potential generated on the inner mitochondrial membrane, their concentrations in mitochondria would increase several orders of magnitude in comparison with the initial concentrations. Due to a high distribution coefficient in the membrane, their concentration increased additionally by four orders of magnitude. Structural similarity to ubiquinone or plastoquinone, natural components of the electron-transporting chains, allows for the regeneration of these agents, making them unique. The mitochondria-targeted antioxidant MitoQ containing the quinone moiety inhibited memory loss, neuropathology, and extended the lifespan in aged 3xTg-AD mice [78]. SkQ1, containing plastoquinone moiety with stronger antioxidant activity, preserved hippocampal neuronal integrity in senescence-accelerated OXYS rats, improved mitochondrial parameters, reduced Aβ levels and tau hyperphosphorylation, promoted neurogenesis and cell survival, prevented synaptic pathology, and improved cognitive function in vivo [79,80,81,82]. Although several mitochondria-addressed drugs have been applied for AD clinical trials, no mitochondria-targeted drug has been approved by the U.S. Food and Drug Administration (FDA). 

Mitochondria, being highly dynamic organelles, continuously fuse and divide and these mitochondrial dynamics are essential for mitochondrial integrity and cell function [61]. The mitochondrial fission is controlled by dynamin-related protein Drp1, mitochondrial fission factor (MFF), and Fission-1 (Fis1). Mitofusins 1 (Mfn1) and 2 (Mfn2) and optic atrophy 1 (OPA1) are proteins responsible for the fusion pathway. 

Intense research for several decades has revealed that dynamin-related protein Drp 1, the evolutionarily highly conserved protein, is critical for mitochondrial division, size, shape, and distribution throughout the neuron. Its structure and mechanism of action are well-studied [83,84]. 

Aβ was found to interact with Drp, triggering the increase in free radical production, which in turn activated Drp1 and Fis, causing excessive mitochondria fragmentation, the defective transport of mitochondria to the synapses, lowered synaptic ATP production, and ultimately, synaptic dysfunction. The protein P-tau can also interact with Drp1, enhancing its activity and leading to excessive fragmentation of mitochondria and mitochondrial dysfunction in AD [34]. Importantly, excessive mitochondrial fragmentation such as a mitochondrial bioenergetic deficit was detected as a prominent early event, contributing to mitochondrial dysfunction, synaptic failure, and neuronal cell death and preceding AD pathology in transgenic animal models [85,86]. 

Several attempts have been made to improve the mitochondrial dynamics in AD by reducing Drp 1 activity. A partial deficiency of Drp1 by using a genetic approach that involved crossing heterozygote knockout Drp1(+/−) mice with transgenic APP mice (Tg2576 strain or by crossing Drp1+/− mice with transgenic Tau mice (P301L line)) and creating double mutants (APP × Drp1+/−) or (Tau × Drp1+/−) mice protected against Aβ-induced mitochondrial and synaptic toxicities in AD neurons [87,88]. 

Mdivi-1, a cell-permeable inhibitor of Drp1 GTPase activity blocking the self-assembly and polymerization of Drp1 [89], enhanced the mitochondrial fusion activity, lowered fission machinery, and increased biogenesis of synaptic proteins in N2a cells affected by Aβ42 [86]. However, the observed beneficial effects of the Mdivi 1-dependent inhibition of Drp1 are challenged by a study showing that Mdivi-1 reversibly inhibits mitochondrial complex I, the main intracellular source of ROS, instead of acting as a specific Drp1 GTPase inhibitor [90]. 

Moreover, the use of Mdivi-1 has no prospects for long-term use, since mitochondrial fission and fusion are absolutely necessary for maintaining mitochondrial homeostasis (see above), which are an integral part of mitochondrial quality and quantity control. 

In this regard, mitochondrial-targeted antioxidants deserve to be considered as promising drugs (therapeutic agents) for combating AD, and there are several reasons for assumption.

Recently, we have shown that very low concentrations of the prooxidant t-BHP, not yet causing oxidative stress discernible with fluorescent probes, triggered the mitochondrial fragmentation in yeast cells, which was not only prevented, but even totally reversed upon treatment by mitochondrial-targeted antioxidants of the SkQ family [61,91]. These findings imply that Drp1p mediated mitochondrial fission originates largely from mitochondrial ROS (mROS) and might be a very sensitive sensor of them. Moreover, by using giant *Dipodascus magnusii* cells normally containing a branched mitochondrial reticulum, it was clearly shown that the prooxidant-induced oxidative stress initially developed only in mitochondria far preceded the appearance generalized oxidative stress in the yeast cell. The preincubation of cells with the mitochondria-targeted antioxidant SkQ1 substantially diminished mROS. Most importantly, the mitochondrial fragmentation induced by mROS preceded the development of the generalized oxidative stress [92]. The main rather optimistic conclusions can be drawn from these and the above described results. Mitochondrial fragmentation is not only the prominent early event preceding AD pathology [85], but it can be a hallmark, a biomarker of the earliest preclinical stage of AD with an effective therapy, based on mitochondria targeted lipophilic antioxidants.

## 4. Materials and Methods

### 4.1. Chemical Reagents

Bacto agar, Bacto peptone, Bacto yeast extract, dithiothreitol (DTT), and Tris (ultra-pure) were purchased from Becton, Dickinson, and Company (Franklin Lakes, NJ, USA); ADP, ampicillin, Anti-Rabbit IgG Peroxidase antibody, antimycin A, ATP, 3-amino-1,2,4-triazole, carbonyl cyanide *m*-chlorophenylhydrazon (CCCP), DABCO, P5-di(adenosine-5)pentaphosphate (Ap5A), EDTA, EGTA, fatty acid-free BSA, glucose, glucose-6-phosphate dehydrogenase, LiAc, mannitol, MgCl_2_, Mowiol 4-88, NaCl, NADP, (NH_4_)_2_SO_4_, oligomycin, Phenol Red, phosphoenolpyruvate, pyruvate kinase, rotenone, succinic acid, and *tert*-butyl hydroperoxide were from Sigma-Aldrich (St. Louis, MO, USA); Coomassie G-250 and zymolyase were from MP Biomedicals (Santa Ana, CA, USA); CaCl_2_, K_2_HPO_4_, KCl, KH_2_PO_4_, NaCl, and safranin O were from Merck (Darmstadt, Germany); 10× DNA Loading Dye, 10× G+ buffer, 10× O+ buffer, 10× R+ buffer, BSA, dihydroetidium, Sytox Green Dead Cell Stain, DMSO, Gene Jet Gel Extraction Kit, Gene Jet Plasmid Miniprep Kit, Gene Ruler 100 bp+, Gene Ruler 1 kb, Glycogen, Mitotracker Red CmxRos, NotI restriction endonuclease, PageRuler™ Prestained Protein Ladder, Phusion High-Fidelity PCR Kit, PvuII restriction endonuclease, Rapid DNA Ligation Kit, RNAse-A, SuperSignal™ West Dura Extended Duration Substrate, and XhoI restriction endonuclease were from Thermo Fisher Scientific (Waltham, MA, USA); Agar, agarose LE2, ethidium bromide, and glycerol (ultra-pure) were from Helicon (Moscow, Russian Federation); LB BROTH Miller (Luria–Bertani) and NaOAc were from Amresco (Dallas, TX, USA); BbsI (BpiI) restriction endonuclease was from New England Biolabs (Ipswich, MA, USA); sorbitol was from Dia-M (Moscow, Russia); oligonucleotides were from DNA-Synthesis (Moscow, Russia). SkQThy (10-(2-isopropyl-5-methyl-1,4-benzoquinonyl-6)-decyl (triphenyl)phosphonium bromide) was kindly provided by Dr. D.S. Esipov from the Belozersky Research Institute of Physico-Chemical Biology MSU, Moscow, Russia.

### 4.2. Cell Cultures

*Escherichia coli*, strain XL1-Blue (Evrogen, Russia), was used for plasmid propagation. 

The *Y. lipolytica* yeast, strain Po1f, was obtained from the National Bioresource Center—All-Russian Collection of Microorganisms (RCM, Moscow, Russia). All the strains used in this study are listed in Table 1. Cells were as described in [92].

### 4.3. Plasmid and Yeast Strain Construction

The primer design was based on the nucleotide sequences of the genes encoding Aβ42, Aβ40, and eGFP, respectively, so that the sequences of PCR products consisted of the full-length Aβ42 and Aβ40 eGFP nucleotide sequences.

The Aβ42 and Aβ40 open reading frames were amplified from synthesized oligonucleotides (DNA-Synthesis, Moscow, Russia) by using pairs of primers, namely, Aβ42-BbsI-Fw1/Aβ42-BbsI-Rev1 for the pZ-Aβ42 construction, Aβ42-BbsI-Fw1/Aβ40-BbsI-Rev1 for the pZ-Aβ40 construction, Aβ42-BbsI-Fw3/Aβ42-BbsI-Rev1 for pZ-eGFP-Aβ42 construction, and Aβ42-BbsI-Fw3/Aβ40-BbsI-rev1 for pZ-eGFP-Aβ40 construction.

The eGFP open reading frame was amplified from pUC:FCP:ShBle:FCP:EGFP (Addgene, Watertown, MA, USA) by using pairs of primers, namely, eGFP-BbsI-Fw1/eGFP-BbsI-Rev1 for the pZ-eGFP construction, and eGFP-BbsI-Fw1/eGFP-BbsI-Rev3 for the pZ-eGFP-Aβ42 and pZ-eGFP-Aβ40 constructions.

The primer sequences used in this study are listed in Table 2. The absence of BbsI restriction sites in the target gene sequences was determined, and the lack of autocomplementarity of the ends of the PCR products was estimated using SnapGene software (GSL Biotech LLC, San Diego, CA, USA).

PCR was carried out as described in [92].

To create the target constructs, the pZ-express++ plasmid with a hybrid hp4d promoter dependent on the growth phase and a ZETA transposon sequence with multiple homology in the *Y. lipolytica* genome was chosen, which ensures a high copy number of the plasmid during recombination and, as a result, a high level of expression of the target protein. The plasmid also has an ampicillin-resistance gene and a prototrophic factor for uracil URA3 from the *Y. lipolytica* genome. Insertion of the PCR products into the pZ-express++ vector was performed as described in [92]. The sequences of Aβ42, Aβ40, eGFP-Aβ42, eGFP-Aβ40, and eGFP were inserted into the vector. All plasmid constructs were verified by restriction enzyme mapping and DNA sequencing of the inserted fragments. The pZ-express++ plasmid was kindly provided by Dr. Laptev I.A. from the Federal Institution “State Research Institute of Genetics and Selection of Industrial Microorganisms of the National Research Center, Kurchatov Institute, Moscow, Russian Federation.

The transfection of *Y. lipolytica* Po1f cells by electroporation was carried out as described in [92].

### 4.4. Mitochondria Visualization in Y. lipolytica Cells by Structural Illumination Microscopy (SIM)

Visualization of mitochondria in *Y. lipolytica* cells by SIM was performed as described in [92].

### 4.5. Mitochondria Visualization in Y. lipolytica Cells by Widefield Fluorescent Microscopy

For mitochondria staining, *Y. lipolytica* cells were loaded with 200 nM MitoTracker Red CmxRos for 30 min in 50 mM PBS, pH 5.5. The stained cells were plated on a 96-well microscopy plate. Serial optical sections were acquired using an inverted motorized microscope Eclipse Ti2 (Nikon, Tokyo, Japan) with a Perfect Focus autofocusing system, equipped with a 100× CFI Plan Apo Vc Oil objective (NA 1.4), 488- and 561-nm diode light source, and air-cooled CMOS camera DS-Qi2 (Nikon, Tokyo, Japan) under the control of NIS-Elements software (Nikon, Tokyo, Japan). Deconvolution was performed by the Richardson–Lucy algorithm included in the NIS-Elements package. The 3D reconstruction was performed by using Icy software, v. 2.0.2.0 [93].

### 4.6. Assessment of Oxidative Stress and Cell Viability of Y. lipolytica Cells

ROS production was determined with dihydroethidium, and yeast cell viability was detected with Sytox Green Dead Cell Stain [94], as described in [92]. Where indicated, the cells were preincubated for 1 h with 250 nM SkQThy, a mitochondria-targeted consisting of a lipophilic cation triphenylphosphonium linked via a C 10 aliphatic chain with an antioxidant thymoquinone (Thy) with versatile healing abilities [61]. 

### 4.7. Isolation of Y. lipolytica Mitochondria

Mitochondria were isolated by the method developed in our laboratory [64]. The quality of isolated mitochondrial preparations was judged as recommended in [95].

### 4.8. Monitoring of Oxygen Consumption by Yeast Mitochondria

Oxygen consumption by yeast mitochondria was monitored amperometrically using a closed Clark-type oxygen electrode in a continuously stirred, thermostatically controlled 1 mL cell as described in [92]. Respiratory rates of mitochondria were expressed as ng-atoms O/min/mg protein.

### 4.9. Assay of ATP Synthesis by Mitochondria

The ATP synthesis was assayed spectrophotometrically with a DU-650 spectrophotometer (Beckman Coulter, Brea, CA, USA) at 557/618 nm using Phenol Red (a pH-indicating dye), as described in [92].

### 4.10. Assessment of Hydrogen Peroxide Production by Mitochondria

The production of hydrogen peroxide by mitochondria was determined fluorometrically by measuring the oxidation of Amplex Red to resorufin coupled with the enzymatic reduction of hydrogen peroxide by horseradish peroxidase as described in [92]. The basic incubation medium was supplemented with 6 mM aminotriazole (an inhibitor of catalase) and mitochondria (0.2 mg protein/mL).

### 4.11. Mitochondrial Protein Assay

Mitochondrial protein was determined using the Bradford method [96] with BSA as the standard.

### 4.12. Statistical Analysis

All experiments with yeast mitochondria were performed at least three times. For analysis of the mitochondrial morphology, at least fifty cells were examined in each trial. Statistical analyses were performed using one-way ANOVA with the post hoc Tukey HSD test. Data were presented as the mean ± S.E. from at least three independent replicates.

## 5. Conclusions

The only AD treatments approved by the U.S. Food and Drug Administration (USFDA) include the cholinesterase inhibitors donepezil, galantamine, and rivastigmine; they showed, however, only limited beneficial effects in clinical trials and had gastrointestinal side-effects. A genetic snipping technique switching APOE4, the strongest risk factor in AD, to APOE3 or APOE2 isoforms, was successful in mouse models, but its utility to clinical trials has proven challenging [97].

Several developed neurotheranostic nanosystems [98] such as phenothiazine derivatives marked by a high affinity for the amyloid plaque in the brain of transgenic mice also serve as a near infra-red fluorescent diagnostic tool [99]; quantum dots, having unique optical and electronic properties, and hence useful in the imaging and diagnostics of AD biomarkers as well as promising high-technologies for better AD diagnostics and therapy, may only be applied in the future [100,101]. 

Currently, AD treatments are limited to only symptomatic management. When a situation seems hopeless, it is useful to remember the past. The Nigella sativa plant, also known as black seed, was regarded in the Middle East as a part of an overall holistic approach to health and named “Panacea” (in Latin, meaning “cure all”) and Habbat el Baraka (in Arabic, translated as “Seeds of Blessing”). Thymoquinone (TQ), the constituent of SkQThy with strong antioxidant activity, is the essential and the most pharmacologically active component of N. sativa with versatile pharmacological properties including anti-inflammatory, anti-arthritis, immunomodulatory, anti-diabetic, anti-asthmatic, anti-hypertensive, anti-sepsis, anti-encephalomyelitis, anti-pancreatitis, anti-parkinsonism, anti-cancer, anti-angiogenic, anti-tuberculosis, antidepressant, anxiolytic, analgesic, and antipsychotic (reviewed by [61]). The problems with the bioavailability of TQ were overcome by the targeted delivery of TQ-derivatives to mitochondria. Mitochondrial-targeted agents mitigated the development of many diseases including AD (see above), thus contributing to a novel area of therapy called mitotherapy [102].

We hope that collectively, long-term multinutrient interventions including effective mitochondria-targeted antioxidants as an important element of mitotherapy and the prevention of modifiable risk factors related to lifestyle [103] along with caloric restriction and aerobic exercises that slow down the aging process, prolong the health period, improve memory and cognition in model organisms, will have good potential for the retarding of AD and dementia. Simple yeast AD models can be promising in searching for substances for treating the disease.

## Figures and Tables

**Figure 1 ijms-24-00900-f001:**
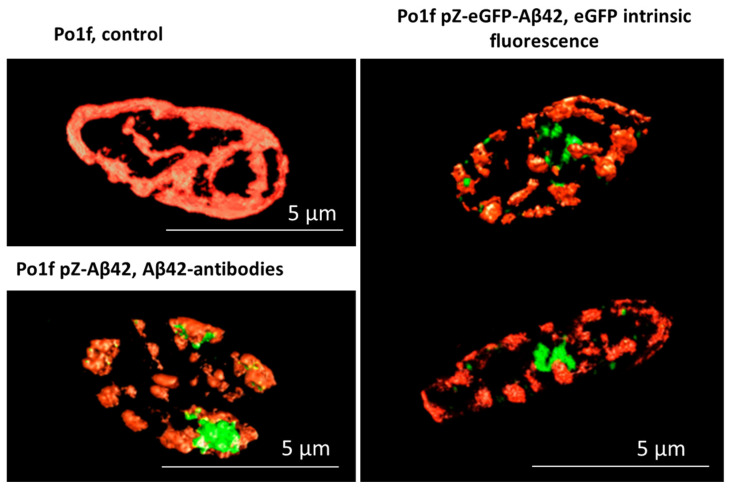
Structure of mitochondria in the *Y. lipolytica* mutants. Po1f (the control, (**top left panel)**), pZ-Aβ42 (**bottom left panel**), and pZ-eGFP-Aβ42 (**right panel**) cells were loaded with 500 nM MitoTracker Red CmxRos (visualizing mitochondria, shown in red) for 30 min. Aβ42 aggregates were visualized with eGFP intrinsic green fluorescence (**right panel**) and antibody treatment (**bottom left panel**).

**Figure 2 ijms-24-00900-f002:**
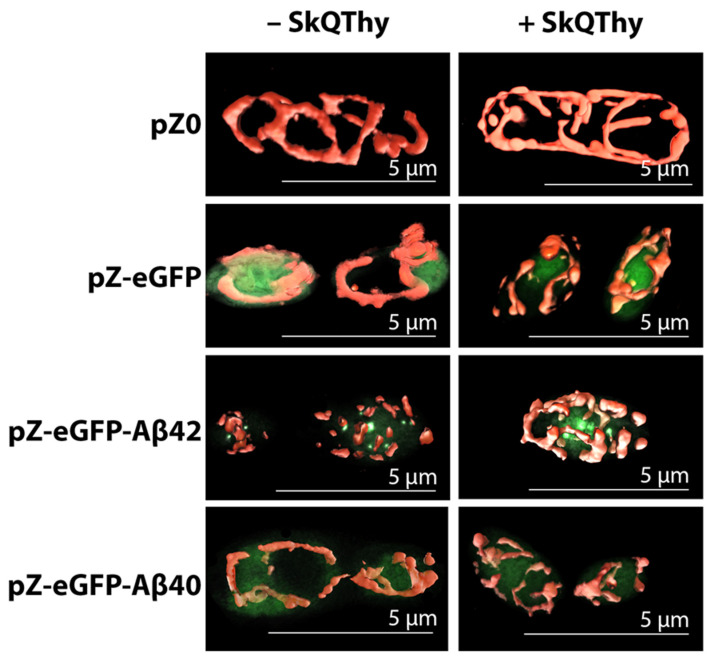
Structure of mitochondria in the *Y. lipolytica* mutants. Effect of SkQThy. Aβ42 aggregates were visualized with eGFP intrinsic fluorescence (shown in green). Cells were preincubated without (**left panel**) or with 250 nM SkQThy (**right panel**) for 1 h and stained with 500 nM MitoTracker Red for 30 min.

**Figure 3 ijms-24-00900-f003:**
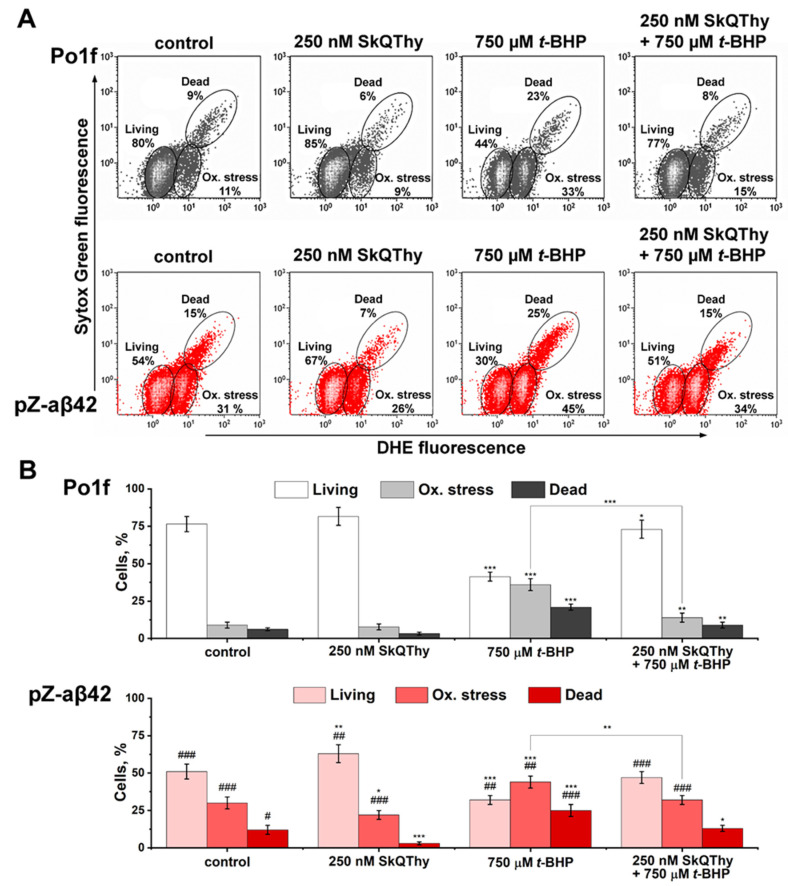
Oxidative stress and cell death in the *Y. lipolytica* strains Po1f (the control, top panel) and pZ-Aβ42 cells (bottom panel). Effects of the prooxidant t-BHP and antioxidant SkQThy. (**A**) Flow cytometry measurements. The results of one representative experiment are presented. (**B**) Histograms of the results obtained by the low cytometry measurements from three independent experiments. Statistical analyses were carried out by the one-way ANOVA test. Symbols: * marks differences between the results obtained for the Po1f strain under different conditions; *** *p* < 0.001, ** 0.001 < *p* < 0.01, * 0.01 < *p* < 0.05; # marks differences between the Po1f and pZ-Aβ42 samples; ### *p* < 0.001, ## 0.001 < *p* < 0.01, # 0.01 < *p* < 0.05.

**Figure 4 ijms-24-00900-f004:**
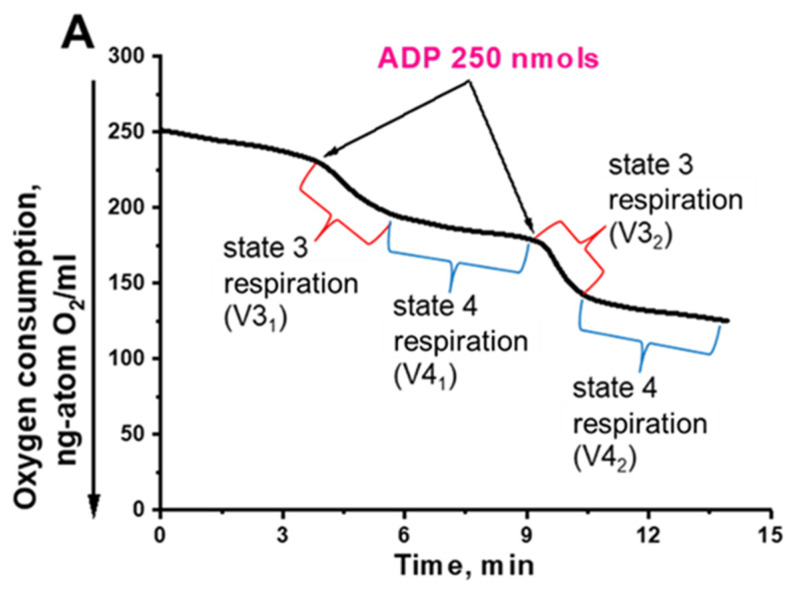
Amperometric recording of oxygen consumption by the isolated *Y. lipolytica* mitochondria. (**A**) *Y. lipolytica* Po1f; (**B**) *Y. lipolytica* Po1f pZ-Aβ42. The basic incubation medium containing 0.6 M mannitol, 2 mM Tris-phosphate, pH 7.2, 1 mM EDTA was supplemented with 20 mM Tris-pyruvate + 5 mM Tris-malate and mitochondria (0.5 mg protein/mL). Where indicated, 250 nmols of ADP were added. Respiratory control ratios upon successive ADP additives were: (**A**) 4.0, 6.8; (**B**) 3.2, 3.3.

**Figure 5 ijms-24-00900-f005:**
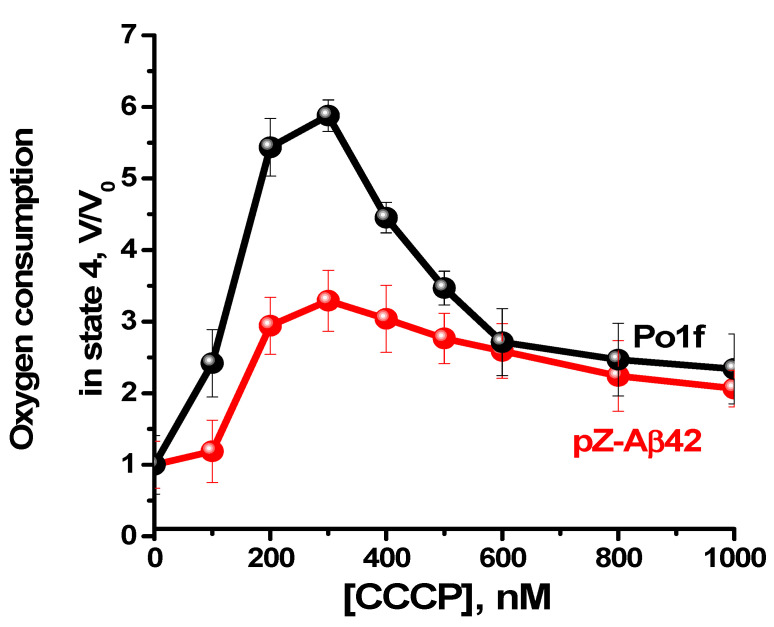
Oxygen consumption by *Y. lipolytica* mitochondria. The incubation medium was supplemented with 20 mM Tris-pyruvate + 5 mM Tris-malate and mitochondria (0.2 mg protein/mL).

**Figure 6 ijms-24-00900-f006:**
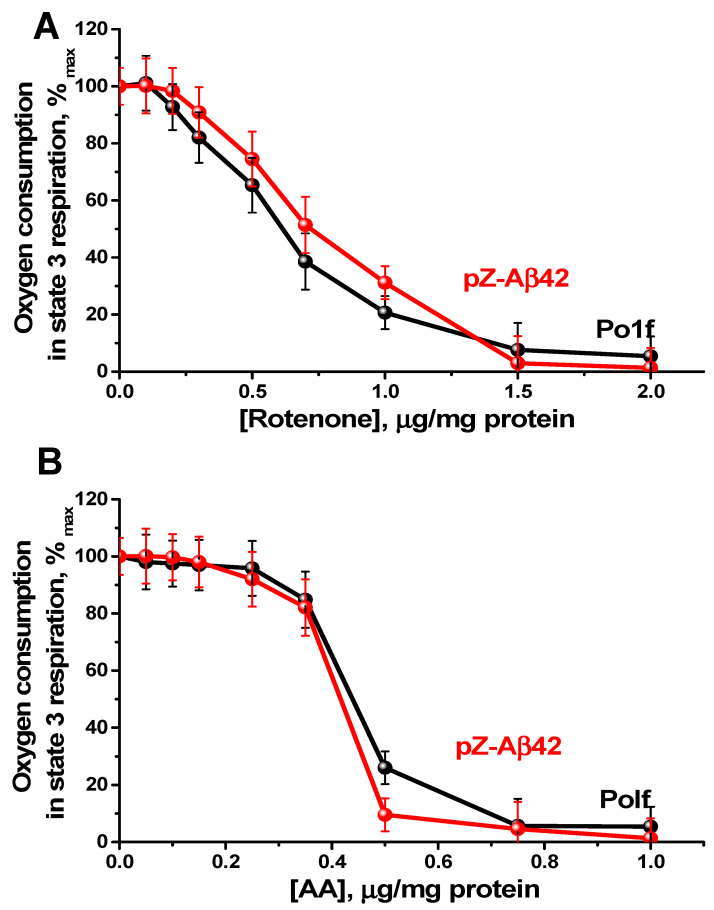
Inhibition of oxygen consumption by *Y. lipolytica* mitochondria in state 3 respiration by respiratory inhibitors: rotenone (**A**) and antimycin (**B**). Incubation medium was supplemented with 20 mM Tris-pyruvate + 5 mM Tris-malate, 1 mM ADP, and mitochondria (0.5 mg protein/mL).

**Figure 7 ijms-24-00900-f007:**
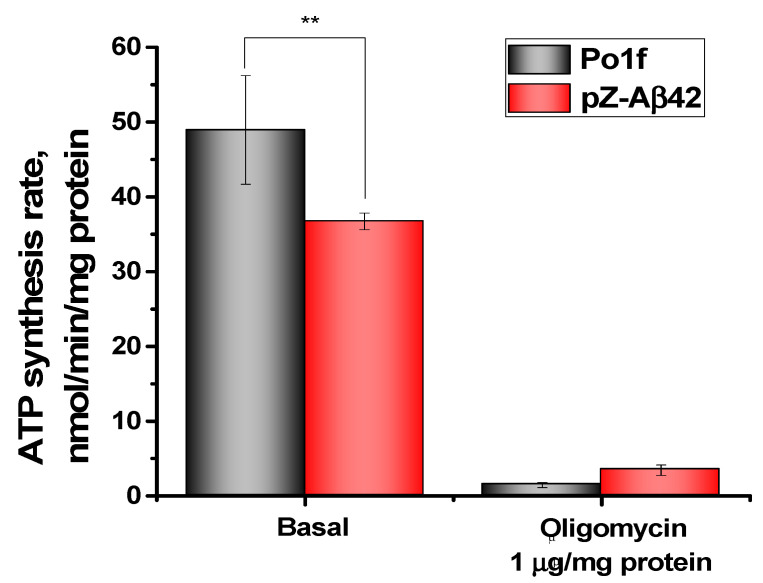
ATP production by *Y. lipolytica* mitochondria. The basic incubation medium was supplemented with 20 mM Tris-pyruvate + 5 mM Tris-malate, 5 µM Phenol Red, 6 μM Ap5A, and mitochondria (0.2 mg protein/mL). The statistical analyses were carried out by one-way ANOVA. ** 0.001 < *p* < 0.01.

**Figure 8 ijms-24-00900-f008:**
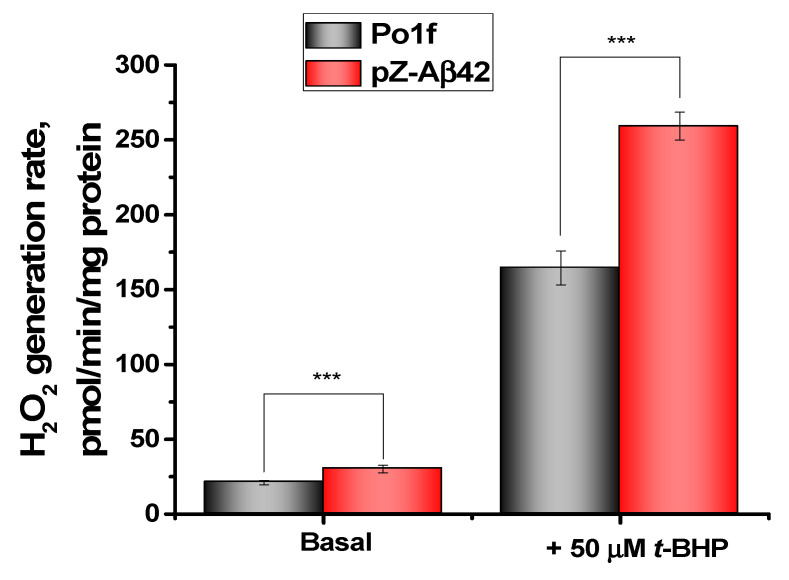
Hydrogen peroxide generation by *Y. lipolytica* mitochondria. The basic incubation medium was supplemented with 20 mM Tris-pyruvate + 5 mM Tris-malate, 6 mM aminotriazole, 9 U/mL horseradish peroxidase, 5 μM Amplex Red, and mitochondria (0.2 mg protein/mL). Statistical analyses were carried out by one-way ANOVA. *** *p* < 0.001.

**Table 1 ijms-24-00900-t001:** List of strains.

Strain	Description
Po1f	MatA, leu2-270, ura3-302, xpr2-322, axp-2
pZ-0	Po1f + pZ URA3, xpr2Δ
pZ-eGFP	Po1f + pZ-eGFP URA3 xpr2Δ
pZ-Aβ42	Po1f + pZ-Aβ42 URA3 xpr2Δ
pZ-eGFP-Aβ42	Po1f + pZ-eGFP-Aβ42 URA3 xpr2Δ
pZ-Aβ40	Po1f + pZ-Aβ40 URA3 xpr2Δ
pZ-eGFP-Aβ40	Po1f + pZ-eGFP-Aβ40 URA3 xpr2Δ

**Table 2 ijms-24-00900-t002:** List of primers.

Primer	Sequence
Aβ42-BbsI-Fw1	TAGAAGACGCAATGGATGCGGAATTTCGC
Aβ42-BbsI-Rev1	TAGAAGACGCGCGCTCACGCAATCACCACG
Aβ42-BbsI-Fw3	TAGAAGACATCAAGATGGATGCGGAATTTCGC
Aβ40-BbsI-Rev1	TAGAAGACGCGCGCTCACACCACGCCGCC
eGFP-BbsI-Fw1	TAGAAGACTAAATGGTGAGCAAGGGCGAGGAG
eGFP-BbsI-Rev1	TAGAAGACGCGCGCTTACTTGTACAGCTCGTCCATG
eGFP-BbsI-Rev3	TAGAAGACCGCTTGTACAGCTCGTCCATGC

## Data Availability

The data used to support the findings of this study are available from the corresponding author upon request.

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
