# Peer review of "Altered Mitochondrial Morphology and Bioenergetics in a New Yeast Model Expressing Aβ42"

_ijms, 2023, doi:10.3390/ijms24020900_

Round 1

Reviewer 1 Report

The manuscript of “Altered mitochondrial morphology and bioenergetics in a new yeast model expressing Aβ42” by Khoren K. Epremyan and co-authors aims to study the impact of the β-amyloid peptide Aβ42 (the main biomarker of of Alzheimer's disease (AD)) on the ultrastructure and function of mitochondria in the genetic yeast model of AD. The authors have significant background in the field of research of yeast models to identify mitochondrial dysfunction. In the present work, the authors constructed and examined the genetically modified Yarrowia lipolytica, a non-toxic aerobic “multitalented” yeast species, expressing Aβ42, eGFP-Aβ42, Aβ40, and eGFP-Aβ40. The advantage of the approach applied is that the Po1f strain of Y. lipolytica used in the work is auxotrophic for uracil and leucine and has a deletion of the Xpr2 gene encoding an extracellular protease, which makes it possible to grow mutants on selective media. Using cutting-edge techniques of structural illumination and widefield fluorescent microscopies for mitochondrial visualization, the authors demonstrated that eGFP-Aβ42 cells possessed mitochondrial disruption and fragmentation, while the yeast cells expressing Aβ40 retained the mitochondrial reticulum and were identical to the control cells expressing only GFP. In addition, the authors conducted a comprehensive assessment of the functional activity of mitochondria from the yeast strains, including measurements of mitochondrial respiration and oxidative phosphorylation, ATP synthesis, hydrogen peroxide production and resistance to the induction of oxidative stress. It was fist found that submicromolar (250 nM) concentrations of SkQThy, the most effective mitochondrial-directed lipophilic antioxidant in the SkQs-family, substantially prevented mitochondrial fragmentation and reduced oxidative stress in Y. lipolytica Po1f pZ-Aβ42. These results are in accordance with the data previously obtained by the authors that the mitochondrial fragmentation was mainly induced by mitochondria-derived ROS and preceded the development of the generalized oxidative stress.

The manuscript is very interesting and well written; all the conclusions are supported by the data obtained. The topic of the manuscript is highly relevant and timely in view of recent the statistics on the incidence of AD in the world. Since there are currently no effective treatments for AD, the use of antioxidants targeting mitochondria may be a promising approach to mitigate or even prevent the harmful effects of the disease. The manuscript contributes to the development and systematization of knowledge about a novel area of therapy called mitotherapy. The Y. lipolytica model developed by the authors provides a promising platform for rapid screening of mitochondria-targeted drugs.

Minor comments.

1.              In the Introduction section, it would be better to more specify the aim of this study. Some description of the mitochondria-targeted antioxidant SkQThy should be added (or moved from the Conclusion section). The word “eye” should be replaced by “ocular biomarkers” in the sentence: “The newer models for early diagnosis of AD are suggested to use the saliva [44], the eye [45] and the blood [37,46].”

2.              Throughout the text there are some references to the Introduction or Methods sections that would be better omitted. For example, “For comparative analysis of mitochondrial morphology, we used another super-resolution method – widefield fluorescence microscopy (see, Methods)”…. “Since the data in Figures 1 and 2 are fully consistent with those from other models (see Introduction)...”

3.              The titles of Figures 3a and 3b (in the first part) overlap significantly, and therefore the Figures should be joined.

4.              Figure 4a contains some grammatical errors in the labels.

5.              In the Discussion section, pretty much attention is paid to the effects of Mdivi-1, a derivative of quinazolinone, namely, 3-(2,4-dichloro-5-methoxyphenyl)-2-thioxoquinazoline-4-one, cell-permeable inhibitor of Drp1 GTPase activity. At the same time, recent data suggest that Aβ42 plays a role in disrupting proteome responses for signaling, bioenergetics, and morphology in mitochondria (doi: 10.3390/cells10092380). The authors could add a more detailed description of the possible mechanisms of action of Aβ42 on mitochondrial function and ROS production.

Author Response

First of all, we would like to thank the referee for useful comments and advises and are ready to answer the questions raised.

Minor comments.

  1. “In the Introduction section, it would be better to more specify the aim of this study.

Some description of the mitochondria-targeted antioxidant SkQThy should be added (or moved from the Conclusion section).

The word “eye” should be replaced by “ocular biomarkers” in the sentence: “The newer models for early diagnosis of AD are suggested to use the saliva [44], the eye [45] and the blood [37,46].”

Response

We added “Thus, the main goal of the work was to create, to our knowledge, in the first time, an improved Y. lipolytica - based yeast model to detect the direct effect of Aβ42 amyloid expression on the mitochondrial structure and dynamics, the redox status and viability of cells, as well as on the bioenergetics at the mitochondrial level”.

We do believe, however, that the description of SkQthy is most appropriate not in the Introduction, section which is a general description (Overview) of Alzheimer’s disease, but in Discussion section, and especially in Conclusion section, devoted to the search for modern AD diagnostic and therapy where, in our opinion, mitotherapy should take a prominent place.

“The newer models ” . "The eye" as a model.

  1. “Throughout the text there are some references to the Introduction or Methods sections that would be better omitted. For example, “For comparative analysis of mitochondrial morphology, we used another super-resolution method – widefield fluorescence microscopy (see, Methods)”…. “Since the data in Figures 1 and 2 are fully consistent with those from other models (see Introduction)...”

Response

 All references to Introduction or Methods sections were removed.  

  1. “The titles of Figures 3a and 3b (in the first part) overlap significantly, and therefore the Figures should be joined.”

Response

Figures 3A and 3B were joined

  1. “Figure 4a contains some grammatical errors in the labels”.

Response

      Grammatical errors were corrected

  1. “In the Discussion section, pretty much attention is paid to the effects of Mdivi-1, a derivative of quinazolinone, namely, 3-(2,4-dichloro-5-methoxyphenyl)-2-thioxoquinazoline-4-one, cell-permeable inhibitor of Drp1 GTPase activity. At the same time, recent data suggest that Aβ42 plays a role in disrupting proteome responses for signaling, bioenergetics, and morphology in mitochondria (doi: 10.3390/cells10092380).

The authors could add a more detailed description of the possible mechanisms of action of Aβ42 on mitochondrial function and ROS production.”

Response

Mdivi-1 is mentioned only in connection with the attempts of some authors to reduce the excess mitochondrial fragmentation  and their suggestion to use it as a new efficient therapeutic strategy for AD. We reduced this part of the text and wrote at the end of the paragraph” However, the observed beneficial effects of Mdivi 1-dependent inhibition of Drp1 are challenged by a study showing that mdivi-1 reversibly inhibits mitochondrial complex I, the main intracellular source of ROS instead of acting as a specific Drp1 GTPase inhibitor [94]. Moreover, the use of Mdivi-1 has no prospects for the long-term use, since mitochondrial fission and fusion are absolutely necessary for maintaining mitochondrial homeostasis (see, above); they are an integral part of mitochondrial quality and quantity control.”

As to “more detailed description of the possible mechanisms of action of Aβ42 on mitochondrial function and ROS production”, is the subject of our further research using the developed model.  

Reviewer 2 Report

The authors present the study entitled "Altered mitochondrial morphology and bioenergetics in a new yeast model expressing Aβ42".

Here the authors construct a yeast culture model to examine the effects of Aß42 on mitochondria and thus the energy levels and oxidative stress level of the yeast cell.

The study does not present much novel insights of the negativ effects of Amyloid beta 42, neither the authors present any mechanistical links. 

Instead the authors present the effects of amyloid beta on their specific yeast model.

In my opinion the manuscript is missing some verification information which should be presented in this short study showing their yeast model.

1) Did the authors sequence the yeast cells?

2) Did the authors quantify the amount of amyloid beta their model produces? If not the authors should try to quantify the amount e.g. via WB/Elisa. Are the presented effects realistic or is it because the amount Aß42 is low/high concentrated. How can the authors & readers compare the effects from their model with e.g. neuroblastoma cell-line experiments.

3) Figure 4 seems to differ in quality to the other figures. Also theres a misspelling "raspiration" instead of "respiration".

4) In my opinion the discussion is to long and should be shortend. Some parts do not discuss the main topic of the manuscript.

5) "Unless otherwise specified, all experiments with yeast mitochondria were performed at least three times with consistent results." What does "with consistent results." means? This sentence should be removed. The authors should present their raw data and mean+SE.  Statistical analysis will tell the reader if the experiments were done with consistent results.

In my opinion the manuscript is interesting to read and in general well written. 

Author Response

First of all, we would like to thank the referee for useful comments and advises and are ready to answer the questions and criticism raised.

“The study does not present much novel insights of the negative effects of Amyloid beta 42, neither the authors present any mechanistical links.” 

Response

The main goal of the work was to create, to our knowledge, in the first time, an improved Y. lipolytica - based yeast model to detect the direct effect of Aβ42 amyloid expression on the mitochondrial structure and dynamics, the redox status and viability of cells, as well as on the bioenergetics at the mitochondrial level. Presentation of mechanistical links is the next step in the long-term study of the created model.  You will not deny that creating simplified models of diseases, AD, in particular, is a modern brand, main stream, an attempt to move away from diversity of symptoms and sophisticated cross-talk of many cofactors that impede the searching advanced treatment. And the field of AD research has greatly benefited from the use of these models.

To be fair, you might recognize that you have not seen such clear images showing the influence of Amyloid beta 42 and 40 on the structure of the mitochondria. This is due to advantages of yeasts over other models. Similarly, such quality of mitochondria and such detailed analysis of them you have hardly seen in the literature. This is due to a long experience of working with yeast mitochondria and constant improvement of methods of their isolation.

“Instead the authors present the effects of amyloid beta on their specific yeast model.”

Response

The mechanisms underlying propagation and progression of oxidative stress within the cell are the focus of researchers' interests for more than fifteen years. However, the very specific model chosen for these studies over 15 years, cardiomyocytes with their highly ordered network of mitochondria, which make them difficult, if not impossible, to move, was not the best for studying a cross-talk between propagation of oxidative stress within the cell and mitochondrial dynamics. Using individual giant yeast cells (Dipodascus magnusii) with freely moving mitochondria and a combination of flow cytometry assay with time-lapse microscopy, we followed the spatiotemporal development of prooxidant-induced oxidative stress and showed, to our knowledge in the first time, that mitochondrial fragmentation was driven by mitochondrial ROS and preceded the development of the generalized oxidative stress (Rogov et al., 2021, Antioxidants).

Thus, the yeast model proved not to be useful, but also to be the only possible to solve such a stagnant general fundamental problem.

“In my opinion the manuscript is missing some verification information which should be presented in this short study showing their yeast model.”

  • “Did the authors sequence the yeast cells?”

Response

We did sequencing of plasmids before yeast transfection and did not find any mutations in the genes encoding Aβ42, Aβ40 and eGFP. Successful insertion and expression of linearized plasmid fragments into the yeast genome was assessed using the uracil prototrophic factor, eGFP intrinsic fluorescence, and fluorescence associated with treatment with antibodies to Aβ42.

  • “Did the authors quantify the amount of amyloid beta their model produces? If not the authors should try to quantify the amount e.g. via WB/Elisa. Are the presented effects realistic or is it because the amount Aß42 is low/high concentrated. How can the authors & readers compare the

effects from their model with e.g. neuroblastoma cell-line experiments.”

Response

We did not determine the amount of Aβ42. We corrected the expression level by selecting the optimal conditions for the promoter to work according to the eGFP fluorescence intensity. The main goal of the study was to simplify the model system as much as possible to the level of amyloid-cell-mitochondria, in order to determine whether there is a direct effect of amyloid on mitochondria, regardless of the many factors inherent in more complex systems. At the same time, there was no task in quantitative assessment and comparison with other models. It was important for us to qualitatively assess the effect on mitochondria at such an expression level, when the amount of Aβ42 becomes sufficient for the formation of aggregates.

  • “Figure 4 seems to differ in quality to the other figures. Also there is a misspelling "raspiration" instead of "respiration".”

Response

Grammatical errors were corrected

  • “In my opinion the discussion is too long and should be shortend. Some parts do not discuss the main topic of the manuscript.”

Response

Discussion section was reduced

  • “"Unless otherwise specified, all experiments with yeast mitochondria were performed at least three times with consistent results." What does "with consistent results." means? This sentence should be removed. The authors should present their raw data and mean+SE.  Statistical analysis will tell the reader if the experiments were done with consistent results.“

Response  

It’s a common way of presenting a drawing that can tell the mitochondriologist more than just histograms

Round 2

Reviewer 2 Report

Authors edited all my comments suffieciently.